# A Matched-Pair Analysis after Robotic and Retropubic Radical Prostatectomy: A New Definition of Continence and the Impact of Different Surgical Techniques

**DOI:** 10.3390/cancers14184350

**Published:** 2022-09-07

**Authors:** Nicola d’Altilia, Vito Mancini, Ugo Giovanni Falagario, Leonardo Martino, Michele Di Nauta, Beppe Calò, Francesco Del Giudice, Satvir Basran, Benjamin I. Chung, Angelo Porreca, Lorenzo Bianchi, Riccardo Schiavina, Eugenio Brunocilla, Gian Maria Busetto, Carlo Bettocchi, Pasquale Annese, Luigi Cormio, Giuseppe Carrieri

**Affiliations:** 1Department of Urology and Renal Transplantation, Policlinico Riuniti di Foggia, University of Foggia, 71122 Foggia, Italy; 2Department of Urology, Bonomo Teaching Hospital, 76123 Andria, Italy; 3Department of Urology, Sapienza Rome University, 00185 Rome, Italy; 4Department of Urology, Stanford University School of Medicine, Stanford, CA 94305, USA; 5Oncological Urology, Veneto Institute of Oncology (IOV), Istituto di Ricovero e Cura a Carattere Scientifico (IRCCS), 37138 Padua, Italy; 6Department of Urology, University of Bologna, 40126 Bologna, Italy

**Keywords:** prostate cancer, male incontinence, robotic prostatectomy

## Abstract

**Simple Summary:**

The primary goal of radical prostatectomy is oncologic efficacy, but mitigation of the side effects related to prostatectomy is also important to preserve quality of life in these patients. In recent decades, robotic prostatectomy has become increasingly popular due to advantages related to better visualization of the anatomical structures, surgical precision, reduced blood loss, and shorter hospital stay. However, the available literature shows no difference in oncologic outcomes between robotic and open. Some authors suggest differences in its potential to improve functional outcomes, i.e., erectile function and urinary continence. The first is certainly more common but more accepted than urinary incontinence, considering that some patients already suffer from it before surgery and others are not particularly interested in recovering from it, unlike urinary incontinence, which causes psychophysical changes that are often difficult to resolve.

**Abstract:**

Background: Radical prostatectomy is considered the gold-standard treatment for patients with localized prostate cancer. The literature suggests there is no difference in oncological and functional outcomes between robotic-assisted radical prostatectomy (RARP) and open (RRP). (2) Methods: The aim of this study was to compare continence recovery rates after RARP and RRP measured with 24 h pad weights and the International Consultation on Incontinence Questionnaire—Short Form (ICIQ-SF). After matching the population (1:1), 482 met the inclusion criteria, 241 patients per group. Continent patients with a 24 h pad test showing <20 g of urinary leakage were considered, despite severe incontinence, and categorized as having >200 g of urinary leakage. (3) Results: There was no difference between preoperative data. As for urinary continence (UC) and incontinence (UI) rates, RARP performed significantly better than RRP based on objective and subjective results at all evaluations. Univariable and multivariable Cox Regression Analysis pointed out that the only significant predictors of continence rates were the bilateral nerve sparing technique (1.25 (CI 1.02,1.54), *p* = 0.03) and the robotic surgical approach (1.42 (CI 1.18,1.69) *p* ≤ 0.001). (4) Conclusions: The literature reports different incidences of UC depending on assessment and definition of continence “without pads” or “social continence” based on number of used pads per day. In this, our first evaluation, the advantage of objective measurement through the weight of the 24 h and subjective measurement with the ICIQ-SF questionnaire best demonstrates the difference between the two surgical techniques by enhancing the use of robotic surgery over traditional surgery.

## 1. Introduction

RP is considered the gold-standard treatment for patients with localized prostate cancer (PCa) and a life expectancy of more than 10 years [1]. The main goal of RP is oncological efficacy, but care is taken to preserve the anatomical structures surrounding the prostate to reduce the receipt of side effects, without compromising oncologic outcomes. Over the last twenty years, RARP has gained popularity and is now used on more than half of patients undergoing surgery for PCa [2]. The robotic platform allows an immersive surgical experience with magnified 3D high-definition vision and depth perception. The surgeon can see tissue planes clearly, identify structures, and dissect with a higher level of precision. This is expected to result in better oncological and functional outcomes [3].

However, the available literature suggests that there is no difference in oncological outcomes between RARP and RRP. The robotic approach has undeniably reduced blood loss and length of stay, but questions remain as to its ability to improve functional outcomes, such as erectile function and UC. Erectile dysfunction (ED) is a common side effect and those who experience ED after surgery and are motivated to regain erectile function can rely on several effective non-surgical and surgical treatments. Conversely, UI is less common but can be more psychologically challenging due to the fact that very few patients have underlying pre-RP UI. Due to this impact of de novo post-RP incontinence, all are strongly motivated to regain continence after surgery, but treatment options can often require additional surgical procedures, which can be a daunting prospect.

Prospective randomized trials comparing open and robotic surgery demonstrated comparable results in terms of early (85% vs. 84%, *p* = 0.8) [4] and late (91% vs. 90%, *p* = 0.6) [5] continence recovery following both techniques, whereas systematic reviews and meta-analyses showed better continence recovery at 12 months in favor of RARP (OR: 1.53, *p* = 0.03) (R: 2.84, *p* = 0.002) [6,7].

Therefore, due these above conflicting data, the aim of this study was to compare the continence recovery rates after RARP and RRP using pad weights over a 24 h period and to determine predictive factors for continence recovery.

## 2. Materials and Methods

### 2.1. Eligibility Criteria

The Da Vinci Xi surgical system has been available at our institution since January 2016; since then, RARP has become the preferred approach for RP, with RRP being performed only in the few patients who refuse or are not considered fit for robotic surgery due to their comorbidities.

For the present study, we queried our Internal-Review-Board-approved prospective database on PCa to identify patients that underwent RARP (n = 600), had a minimum follow-up of 12 months, and a matched population of patients that underwent RRP (n = 945) between January 2010 and December 2015. Patients with incomplete clinical data or those who had adjuvant treatment after surgery were excluded. Additionally, the first 100 patients who underwent RARP were excluded due to the possible impact of the learning curve on functional outcomes [8].

According to our policy [9,10,11], all patients had uroflowmetry with evaluation of peak flow rate (Q-max) and a post-void residual urinary volume (PVR). Transrectal ultrasound was used to measure both Pvol and to calculate PSA density.

For RRP, we used the technique described by Walsh [12], whereas for the RARP we used an intraperitoneal access and an anterograde dissection through the space of Retzius [13,14]. Pelvic lymph node dissection was carried out according to Briganti nomogram indications using an extended template as recommended by EAU guidelines [1]. The criteria for neurovascular bundle preservation were identical for both procedures and were carefully balanced against the risk of positive surgical margins [15,16,17]. The vesicourethral anastomosis, always preceded by posterior reconstruction [18], required 6 stitches in RRP, while RARP required 2 barbed running sutures. A drain was always placed at the end of each procedure and all patients underwent a retrograde cystogram prior to bladder catheter removal. 

UC was assessed and discussed by two dedicated “Continence Nurses” before surgery and then 7 days after catheter removal. Patients were seen weekly for the first post-operative month in a formal pelvic floor rehabilitation program and then at 3, 6, and 12 months. During all visits, the ability to perform Kegel exercises was assessed; in addition, data on 24 h pad test and ICIQ-7 questionnaire were recorded.

### 2.2. Statistical Analysis

The primary study objective was to compare the rates of patients continent at 1 week, 1 month, 6 months, and 1 year after the two procedures.

To control for measured potential confounders in the data set, a 1:1 propensity score (PS)-matched population of patients undergoing RP was selected using age at surgery, PSA, Biopsy Gleason Grade Group (GGG), and digital rectal exam (DRE). After PS matching, the quality of covariate balance was checked by assessment of standardized differences. Continuous variables were reported as median and interquartile range and compared by the Mann–Whitney U-test, whereas categorical variables were reported as rates and tested by the Fisher’s exact test or the chi-square test, as appropriate. A Kaplan–Meier curve of the probability of urinary continence recovery following RP over time was created. Finally, uni- and multivariable Cox regression analysis, involving the entire study cohort, was used to evaluate predictors of continence. Statistical analyses were performed using Stata-SE 15 (StataCorp LP, College Station, TX, USA). All tests were 2-sided with a significance level set at *p* < 0.05.

## 3. Results

After propensity score matching of the patient population, 482 met the inclusion criteria, 241 patients per group; Table 1 shows the preoperative characteristics of the study population. Of note, there were no differences in age, DRE, PSA, preoperative IIEF-5 score, IPSS, maximum peak flow rate (Q-max), PVR, Pvol, and biopsy GGG.

As for UC rates (Table 2), RARP performed significantly better than RRP at all time points. The continence rate for RARP and RRP was 58% vs. 47% at 1 week (*p* = 0.018), 82% vs. 61% at 1 month (*p* < 0.0001), 92% vs. 75% at 6 months (*p* < 0.0001), and 94% vs. 80% at 1 year (*p* < 0.0001), respectively.

Further, RARP performed significantly better than RRP in UI (Table 3), since severe leakage (>200 gr/24 h) was 40.9% vs. 15.8% at 6 months (*p* = 0.04) and 39.6% vs. 7.1% at 1 year (*p* = 0.02), in favor of RARP. Furthermore, Table 3 shows the trends in the score on the ICIQ questionnaire, which reflects the trend in the overall severity of urinary incontinence.

Kaplan–Maier curves (Figure 1) confirmed a significant difference in continence rates between techniques (Figure 1a), especially in those who received a bilateral nerve sparing (Figure 1b).

In Table 4, the analysis on the entire study population (n = 482 patients) identified the bilateral nerve-sparing technique and robotic surgical approach as having significant predictor value of continence rates (1.25 (CI 1.02,1.54), *p* = 0.030; 1.42 (CI 1.18,1.69), *p* ≤ 0.001).

## 4. Discussion

This study demonstrates better early and late recovery continence and lower rates of UI favoring RARP. Further, we show an overall benefit of the bilateral nerve-sparing approach as a factor in facilitating urinary recovery. Further, regarding time to continence, our data showed not only better but also faster continence recovery with RARP than with RRP. Indeed, RARP performed better than RRP at all evaluation times, considering continence rates were 47% vs. 58% at 1 week (*p* = 0.018), 61% vs. 82% at 1 month (*p* = 0.001), 75% vs. 92 % at 6 months (*p* = 0.018), and 80% vs. 94% at 1 year (*p* = 0.001).

UC rates after open and robotic prostatectomy vary widely in the available literature and several factors may explain these differences. The robotic approach has clear advantages, including better visualization of delicate structures involved in continence recovery [14,15,16,17,18,19,20] and higher surgical precision in the reconstructive phase. However, on the flip side, in reported series, up to 31% might have UI (pad > 0) at 1 year [7]. Patient and cancer characteristics, operative techniques, surgical experience, surgeon dexterity, high-volume centers, post-operative urinary rehabilitation, and adjuvant treatments are all factors to be consider when comparing post-prostatectomy results [21,22]. In this setting, the only existing randomized prospective study showed similar and comparable results in oncological and functional aspects, both in the short and long term [4,5,6,7,21,22], although the functional domain was judged on data acquired from questionnaires administered at follow-up and with no mention of postoperative rehabilitation.

The level of evidence of most of the published data is retrospective and, by nature, limited by selection bias. Furthermore, in the first ten years after the introduction of robotic technology, many studies were characterized by a significant publication bias, with post-RARP continence rates (pad 0) reported up to 50–80%, 90–96%, and 89% to 98%, interesting values after catheter removal compared with open at 6 and 12 months, respectively [23]. Further, Patel et al. [24] reported his technique of RARP provided a 1-year continence rate (pad 0) of 96.4% and Menon et al., a 6-month continence rate (pads ≤ 1) of 92% [13].

Subsequent data have been conflicting in nature regarding any advantages in continence with the robotic approach. An early meta-analysis compared the two techniques in terms of recovery of UC at 12 months and showed significant advantages to a robotic approach (OR: 1.53; 95% CI, 1.04–2.25; *p* = 0.03). Indeed, after RARP, continence recovery ranged from 89% to 100% compared to RRP, which ranges from 80% to 97% [7,25,26,27]. However, this new technology still exposed patients to IU, though with less impact (11% vs. 7%). Similarly, Seo et al. compared 10 matching studies between open and robotic RP with 2214 patients and reported a lower incontinence rate after RARP (RR 0.66, 95% CI 0.45–0.99, I2 = 45%; *p* = 0.040) [28].

Subsequently, several authors began to present data that demonstrated minimal differences in continence between the open and robotic approaches. A 2017 meta-analysis included 78 studies, in which Kun Tang showed no statistical differences in the 3-month and 12-month UC within the two surgical techniques (3 months: OR:1.54; 95% CI: 0.92 to 2.58; *p* = 0.10; 12 months: OR:1.03; 95% CI: 0.84 to 1.27; *p* = 0.75, respectively) [29]. Similarly, Lan Cao et al. published a systematic review and meta-analysis in 2019 with 8522 patients and found no difference in the 12-month continence rates among the two procedures. UC was defined as either no pad use or no leakage (71.6% vs. 70.8%; OR 1.14, 95% CI 0.99 to 1.31, *p* = 0.08) [30]. In a 1:2 matching population series between 294 and 588 patients undergoing RARP and RRP, one-year continence recovery rate (security pad) was overlapping (93.7% vs. 91.8%; *p* = 0.34) also [31]. Several meta-analyses pointed out that the data were insufficient to define a technique as superior, given variables related to surgeon experience, patient history, and oncologic finding.

Quantifying the degree and severity of urinary incontinence has been an area of ongoing debate. Even though pad weight and pad usage are well correlated with quality of life [32], the variability in the definitions used in the literature makes it difficult to objectify an outcome and, therefore, make a real comparison among the results of traditional and technological surgery. The above-mentioned studies in recent years have only assessed incontinence by interviewing the patient based on the number of pads used per day (0, 0–1, one safety pad). Therefore, the differences between the two techniques are not highlighted, emphasizing the use of technology but not for functional advantages.

The rate of functional recovery also varies widely in proportion to pad use. As a result, one study pointed out that the continence rate after RP at 3 and 12 months was 44% and 68% when defined as “pad 0” and increased to 71% and 90% when defined as “pad 0–1” [33]. In addition, a systematic review, reported average values of post-RARP incontinence at 1 year of 16% and 9% according to the definition of “dry” or “a safe pad” [7]. Although Holze et al. argues zero pads best reflects the perception of “dry” [33], Moore et al., suggests a minimum value greater than 4 g/day of leakage as incontinence after prostatectomy, but greater than 20 g/day is characterized as symptomatic incontinence [34]. We, therefore, chose the threshold of a 24 h pad weight test greater than 20 g as a reasonable value for assessing symptomatic male incontinence after an RP. However, a 0 g pad in 24 h is not easy to obtain, as sweat can also increase the weight of the pad and can classify a continent patient as incontinent. For this reason, other definitions are established, such as socially continent. With this, 20 g as a threshold seems reasonable and easy to assess to define a patient as socially continent.

Based on our experience, we have historically chosen to avoid testing UC through interviews about pad count, which is strongly related to subjective feelings of “wetness”. Instead, we choose “third-party” assessment by a urinary specialist nurse who objectively records the 24 h pad weight test at follow-up. We believe it is a reliable tool for assessing the presence and degree of urinary leakage and combining it with those obtained from the recognized ICIQ-7 subjective questionnaire for UI [35].

Others have used 24 h pad weight as an objective measurement of the degree of incontinence. In 2010, Dubbelman was the first to present functional outcomes after RRP using the 24 h pad weight test at 6 months in a series of 66 patients. He reported recovery of continence, defined as 4 g per day, in 44% of patients who were receiving a “Continence Nurses” or folder-guided approach [36]. In 2017, Sathianathen was the first to present functional outcomes after RARP using 24 h pad weight test. Using zero grams as a definition, Sathianathen reported continence recovery after RARP in 20% and 38% of patients, respectively, after 4 and 12 weeks, with average continence recovery at 10 weeks. They also pointed out the fact that if the number of pads had been used as the definition, it would probably have produced better results than UC [37].

However, currently, the 24 h pad test is used by less than 1 out of 10 clinicians [38]. It is considered a controversial way of objectively reporting a measure of male post-prostatectomy UI [39], because the weight of the 24 h pad is directly influenced by daily activity and fluid intake and the same weight can vary by 51 g (95% CI 30.3–72.1, *p* < 0.001) in those who do not change their activity level. On the other hand, what is perceived to be a single wet pad can range from 23 to 121 g and it is these values that make an objective comparison of the degree of leakage possible. More importantly, for each 1-unit increase in pads used, the average pad weight increased by 114.1 g (95% CI 74.8–153.4, *p* < 0.001) [40]. With this in mind, the definition of “Pad 0–1” may not reflect continence status as it may include symptomatic urinary leakage [34]. The added value of our study is to better stratify patients and define already symptomatic incontinent patients with >20 g or more of leakage who, in other definitions, would be continent.

Additional studies suggest that characteristics, such as age, body mass index (BMI), Pvol, and lower urinary tract disorders (LUTS), influence functional outcomes [40,41,42]. As we have previously shown, age did not influence continence recovery in RRP [43], even though Jeong found advantages in young patients, both in early and long-term follow-up [44] and Becker claims that beyond the age of 50, the recovery rate drops from 97.4% to 91.6% [45].

Another interesting insight of the analysis suggests that bilateral sparing of the neurovascular bundles is advantageous in facilitating recovery of UC on an entire study population, especially in RARP, as shown in multivariate analysis (Table 4) and Kaplan–Meier curves (Figure 1b). In this context, the benefits of nerve sparing on continence recovery, both in the short term and long term for RRP and RARP, have already been reported [46,47,48]. These findings are so strong that Steineck continues to recommend nerve sparing, regardless of age and preoperative erectile function, as the benefits are significant [49].

One of the strengths of our study was the management and evaluation of the prostatectomized matching pair population that was entrusted to the “urinary nurse” who trained and collected the data ensuring an unbiased evaluation. The definition was based on the objective measurement of a 24 h pad test in grams rather than on the pad use together with the subjective data obtained from the ICIQ-SF questionnaire. Having stated that, the results obtained justify the use of robotic surgery, not only for less blood loss and hospitalization but also for its impact on the recovery of UC and quality of life.

This work certainly has limitations; first, this is a retrospective analysis of prospectively collected data. Further, body mass index (BMI) and an assessment of preoperative continence were not assessed. However, in the second case, we have a subjective comparison of LUTS through the IPSS questionnaire with overlapping results between the two groups.

## 5. Conclusions

In urologic oncologic surgery, to date, robotic surgery has brought advantages in terms of decreases in blood loss, shorter hospitalization, and, possibly, improved quality of life. When it comes to treating localized prostate cancer, a prostatectomy not only has oncologic but also functional implications. In terms of continence recovery, our study demonstrates that, with a rigorous objective evaluation of postoperative urinary leakage, robotic prostatectomy has advantages over traditional surgery, in both short- and long-term measures of continence.

## Figures and Tables

**Figure 1 cancers-14-04350-f001:**
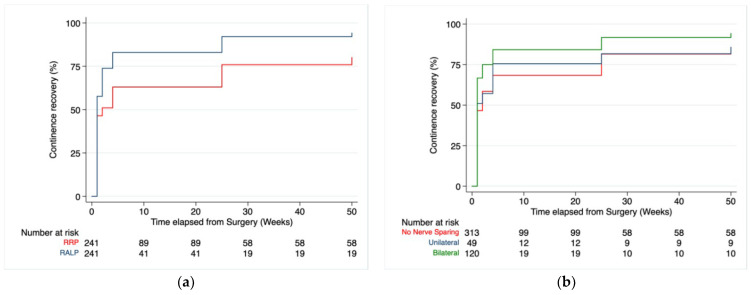
Continence recovery rates between RRP and RARP at follow-up after propensity matching, Pr > chi2 ≤ 0.0001 (**a**). Continence recovery rates divided by nerve sparing grade on the entire population study, Pr > chi2 = 0.0012 (**b**).

**Table 1 cancers-14-04350-t001:** Preoperative characteristics of the study population after propensity matching.

Characteristics	RRP ^1^ n = 241 (50%)	RARP ^1^ n = 241 (50%)	*p*-Value ^2^
**Age at surgery (y)**	66.0 (62.0, 71.0)	66.0 (61.0, 70.0)	0.2
**PSA, (ng/mL)**	6.2 (4.6, 9.1)	6.2 (4.8, 9.2)	0.6
**DRE, n (%)**			
Negative	125 (52%)	119 (49%)	0.6
Positive	116 (48%)	122 (51%)	
**Biopsy GGG, n (%)**			
1	110 (46%)	110 (46%)	0.9
2	57 (24%)	64 (27%)	
3	42 (17%)	41 (17%)	
4	22 (9%)	19 (8%)	
5	10 (4%)	7 (3%)	
**Prostate volume (cm^3^)**	40.0 (31.0, 56.0)	40.0 (32.0, 55.0)	1
**Q-max (mL/s)**	15.6 (12.0, 18.6)	14.0 (11.7, 21.0)	0.7
**PVR (mL)**	20.0 (1.0, 40.0)	30.0 (1.0, 50.0)	0.060
**Pre-operative IPSS**	7.0 (4.0, 12.0)	7.0 (3.0, 12.0)	0.3
**Pre-operative IIEF**	19.0 (11.0, 22.0)	20.0 (15.0, 23.0)	0.067
**Nerve Sparing**			
Non nerve sparing	172 (71.4%)	141 (58.5%)	0.002
Unilateral	26 (10.8%)	23 (9.5%)	
Bilateral	43 (17.8%)	77 (32.0%)	

^1^ Median (IQR); n (%). ^2^ Wilcoxon rank-sum test, Pearson’s Chi-square test.

**Table 2 cancers-14-04350-t002:** Continence recovery rates (pad 0–20 gr) in patients undergoing RRP and RARP at 1 week, 1 month, 6 month, and 1-year follow-up.

Characteristic	RRP ^1^ n = 241 (50%)	RARP ^1^ n = 241 (50%)	*p*-Value ^2^
**1 week**			
Pad < 20 gr, n (%)	114 (47%)	140 (58%)	0.018
ICIQ score *	9.0 (5.0, 13.0)	7.0 (4.0, 10.0)	0.011
**1 month**			
Pad < 20 gr, n (%)	148 (61%)	198 (82%)	<0.0001
ICIQ score *	8 (5, 21)	7 (0, 21.0)	0.001
**6 months**			
Pad < 20 gr, n (%)	180 (75%)	222 (92%)	<0.0001
ICIQ score *	5 (0, 10)	4 (0, 6)	0.029
**1 year**			
Pad < 20 gr, n (%)	193 (80%)	227 (94%)	<0.0001
ICIQ score *	5 (0, 8)	3 (0, 6)	0.09

^1^ Median (IQR); n (%). ^2^ Wilcoxon rank-sum test, Pearson’s Chi-square test. * ICIQ: International Consultation Incontinence Questionnaire—short form (ICIQ-short form).

**Table 3 cancers-14-04350-t003:** Incontinence recovery rates (pad > 20 gr) in patients undergoing RRP and RARP at 1 week, 1 month, 6 month, and 1-year follow-up.

Characteristic	RRP ^1^ n = 241 (50%)	RARP ^1^ n = 241 (50%)	*p*-Value ^2^
**1 Week**	**n = 127**	**n = 101**	
21–200 gr, n (%)	63 (49.6%)	82 (81.2%)	0.00001
ICIQ score *	20.5 (18, 21)	20.5 (18, 21)	0.27
>200 gr, n (%)	64 (50.4%)	19 (18.8%)	0.00001
ICIQ score *	21 (21, 21)	21 (21, 21)	0.72
**1 Month**	**n = 93**	**n = 43**	
21–100 gr, n (%)	44 (47.3%)	35 (81.4%)	0.00001
ICIQ score *	15 (9, 21)	10 (7, 15)	0.27
>200 gr, n (%)	51 (54.8%)	8 (18.6%)	0.00001
ICIQ score *	21 (21, 21)	21 (21, 21)	0.4
**6 Month**	**n = 61**	**n = 19**	
21–100 gr, n (%)	36 (59.1%)	16 (84.2%)	0.04
ICIQ score *	10 (7, 17)	10 (6, 16)	0.52
>200 gr, n (%)	25 (40.9%)	3 (15.8%)	0.04
ICIQ score *	21 (18, 21)	15 (10, 16)	0.58
**1 Year**	**n = 48**	**n = 14**	
21–100 gr, n (%)	29 (60.4%)	13 (82.9%)	0.02
ICIQ score *	10 (8, 15)	10 (7, 10)	0.51
>200 gr, n (%)	19 (39.6%)	1 (7.1%)	0.02
ICIQ score *	21 (21, 21)	21 (21, 21)	0.52

^1^ Median (IQR); n (%). ^2^ Wilcoxon rank-sum test, Pearson’s Chi-square test. * ICIQ: International Consultation Incontinence Questionnaire - short form (ICIQ-short form).

**Table 4 cancers-14-04350-t004:** Univariable and multivariable Cox Regression Analysis to predict continence recovery in the entire population.

	Univariable Cox Regression Analysis			Multivariable Cox Regression Analysis		
Covariate	Haz. Ratio	95% CI	*p* Value	Haz. Ratio	95% CI	*p* Value
**Age**	0.99	0.98, 1.00	0.174			
**GGG biopsy**						
1	Ref.			Ref.		
2	1.05	0.84, 1.31	0.661	1.02	0.82, 1.28	0.857
3	0.87	0.67, 1.13	0.301	0.90	0.69, 1.17	0.430
4	0.88	0.68, 1.14	0.329	0.98	0.75, 1.28	0.876
5	1.03	0.72, 1.47	0.877	1.15	0.80, 1.65	0.462
**Prostate volume**	1.00	0.99, 1.00	0.231			
Q-max	1.00	1.00, 1.01	0.464			
PVR	1.00	1.00, 1.01	0.397			
IPSS	0.99	0.97, 1.00	0.155			
IIEF	1.01	1.00, 1.02	0.163			
**Nerve sparing**						
No	Ref.			Ref.		
Unilateral	1.06	0.79, 1.43	0.711	1.01	0.75, 1.37	0.932
Bilateral	1.35	1.10, 1.64	0.003	1.25	1.02, 1.54	0.030
**Catheters days**						
≥8	Ref.					
<8	0.85	0.72, 1.00	0.056	0.88	0.75, 1.05	0.159
**Surgery type**						
RRP	Ref.			Ref.		
RARP	1.46	1.23, 1.73	<0.001	1.42	1.18, 1.69	<0.001

## Data Availability

The data presented in this study are available on request from the corresponding author. The data are not publicly available.

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
