# Peer review of "A Matched-Pair Analysis after Robotic and Retropubic Radical Prostatectomy: A New Definition of Continence and the Impact of Different Surgical Techniques"

_cancers, 2022, doi:10.3390/cancers14184350_

Round 1

Reviewer 1 Report

I reviewed the manuscript entitled “A match-pair analysis after robotic and retropubic radical prostatectomy: new definition of continence and impact of different 3 surgical techniques” by Nicola d’Altilia, et al.

The study design and results were reasonable and easily communicated to me and other readers. I also agree with them that early continence recovery has a positive impact on quality of life and subsequent recovery. In addition, the current recognition and problem regarding the definition of urinary continence, and their clues to those issues are well documented in the Discussion part.

Therefore, this manuscript is worthy of publication despite some minor grammatical errors.

Author Response

I reviewed the manuscript entitled “A match-pair analysis after robotic and retropubic radical prostatectomy: new definition of continence and impact of different 3 surgical techniques” by Nicola d’Altilia, et al.

The study design and results were reasonable and easily communicated to me and other readers. I also agree with them that early continence recovery has a positive impact on quality of life and subsequent recovery. In addition, the current recognition and problem regarding the definition of urinary continence, and their clues to those issues are well documented in the Discussion part.

Therefore, this manuscript is worthy of publication despite some minor grammatical errors.

We thank you for your time and for the comments. The paper, after the latest reviews, has been further improved.

Reviewer 2 Report

Authors performed a match-paired comparison between open and robotic radical prostatectomy in order to compare post-operative urinary incontinence based on 24-hour pad measurement instead of simple questionnaire response. The use of dedicated nurses is important and strengthens the results due to good methodology of the study. This is a very conflicting issue in urological literature and good quality studies are needed since results can vary according to continence definition. For example a 0 gram 24-hour pad test is difficult to achieve since sweat can also increase pad weight and may categorize a continent patient as incontinent. Therefore other definitions such as socially continent are established. Bearing this in mind, 20 gr as a threshold seams reasonable to define a patient as socially continent and this should be noted within the discussion and maybe also in the title of the manuscript.

Following this, the use of physiotherapy after surgery may alter findings since this practice has shown positive effect on stress urinary incontinence in males. Therefore it should be noted how adherent patients were for example if patients after surgery did not adhere to PFMT at the same rate as patients after robotic surgery this might have affected study findings.

Two more factors not assessed in the study and may affect findings are BMI and pre-operative urinary continence of patients and if no such data are available, authors should notify this as a limitation.

Finally an extensive review of English language is needed since there are several errors and at some points it is difficult to understand the meaning of sentences.

Author Response

Authors performed a match-paired comparison between open and robotic radical prostatectomy in order to compare post-operative urinary incontinence based on 24-hour pad measurement instead of simple questionnaire response. The use of dedicated nurses is important and strengthens the results due to good methodology of the study. This is a very conflicting issue in urological literature and good quality studies are needed since results can vary according to continence definition. For example a 0 gram 24-hour pad test is difficult to achieve since sweat can also increase pad weight and may categorize a continent patient as incontinent. Therefore other definitions such as socially continent are established. Bearing this in mind, 20 gr as a threshold seams reasonable to define a patient as socially continent and this should be noted within the discussion and maybe also in the title of the manuscript.

Thank you for your suggestion. I have modified and inserted your contribution in line 240-143.

Following this, the use of physiotherapy after surgery may alter findings since this practice has shown positive effect on stress urinary incontinence in males. Therefore it should be noted how adherent patients were for example if patients after surgery did not adhere to PFMT at the same rate as patients after robotic surgery this might have affected study findings.

Good point. For this reason, we included in the study only those patients who performed the follow-up at our office and reported performing the exercises at home. Confidentially, we inform you that the patients who did not perform PFMT (and in any case excluded from the study because they had no objective evaluation) are about 30 in the robotic group, and they often waived controls because they reported being continent. In the RRP group, we do not have with certainty data on how many dropped out of the exercises because they were continent or out of resignation, since they did not remember at the interview (it has been more than 7 years). For this reason, we included only those for whom we had objective documentation obtained through office accesses.

Two more factors not assessed in the study and may affect findings are BMI and pre-operative urinary continence of patients and if no such data are available, authors should notify this as a limitation.

The observation was included in the study limits in line 297-300.

Finally an extensive review of English language is needed since there are several errors and at some points it is difficult to understand the meaning of sentences.

An extensive English review has been done by prof. Chung of Stanford University. At this regard he has been added between the authors. We thank for this valuable suggestion, the paper has now another form.

Round 2

Reviewer 2 Report

Dear authors thank you for replying to the comments